# Unpredictable Chemical Diversity of Essential Oils in *Cinnamomum burmanni* (Lauraceae) Living Collections: Beyond Maternally Inherited Phylogenetic Relationships

**DOI:** 10.3390/molecules29061206

**Published:** 2024-03-08

**Authors:** Peiwu Xie, Qiyi Yang, Jielian Chen, Tieyao Tu, Huiming Lian, Boxiang He, Yanling Cai

**Affiliations:** 1Guangdong Province Key Laboratory of Silviculture, Protection and Utilization, Guangdong Academy of Forestry, Guangzhou 510520, China; xiepeiwu@sinogaf.cn (P.X.); chenjielian2016@163.com (J.C.); lhming@sinogaf.cn (H.L.); heboxiang@sinogaf.cn (B.H.); 2State Key Laboratory of Plant Diversity and Specialty Crops/Guangdong Provincial Key Laboratory of Applied Botany/Key Laboratory of National Forestry and Grassland Administration on Plant Conservation and Utilization in Southern China, South China Botanical Garden, Chinese Academy of Sciences, Guangzhou 510650, China; qiyiyang@scbg.ac.cn (Q.Y.); tutieyao@scbg.ac.cn (T.T.); 3South China National Botanical Garden, Guangzhou 510650, China

**Keywords:** *Cinnamomum burmanni*, essential oil, chemotypes, plastid genome, botanical garden

## Abstract

The genus *Cinnamomum* encompasses diverse species with various applications, particularly in traditional medicine and spice production. This study focuses on *Cinnamomum burmanni*, specifically on a high-D-borneol-content chemotype, known as the Meipian Tree, in Guangdong Province, South China. This research explores essential oil diversity, chemotypes, and chloroplast genomic diversity among 28 *C. burmanni* samples collected from botanical gardens. Essential oils were analyzed, and chemotypes classified using GC-MS and statistical methods. Plastome assembly and phylogenetic analysis were conducted to reveal genetic relationships. Results showed distinct chemotypes, including eucalyptol and borneol types, with notable variations in essential oil composition. The chloroplast genome exhibited conserved features, with phylogenetic analysis revealing three major clades. Borneol-rich individuals in clade II suggested a potential maternal inheritance pattern. However, phylogenetic signals revealed that the composition of essential oils is weakly correlated with plastome phylogeny. The study underscores the importance of botanical gardens in preserving genetic and chemical diversity, offering insights for sustainable resource utilization and selective breeding of high-yield mother plants of *C. burmanni*.

## 1. Introduction

The genus *Cinnamomum* s.l. consists of two segregated genera (*Camphora* and *Cinnamomum* s.s.) and ca. 350 species of perineal trees and shrubs distributed in tropical to subtropical Asia [1]. Many species in this genus have a wide range of usages. *Camphora officinarum* Nees ex Wall. (*Ci. camphora*), *Ci. aponicum* Sieb., *Ci. burmanni*, etc. are good ornamental and timber trees [2]. The bark has been widely used as spice, including *Ci. verum* J. Presl (true cinnamon), *Ci. tamala* (Buch.-Ham.) T. Nees & C. H. Eberm (Indian cassia), *Ci. cassia* (L.) J.Presl (Chinese cassia), *Ci. citriodorum* Thwaites (Malabar cinnamon) and *Ci. burmanni* (Nees & T. Nees) Blume (Indonesian cassia) [3]. Moreover, essential oils from cinnamon and camphor plants (*Cinnamomum* s.s. spp. and *Camphora* spp.) exhibit various constituents, which can be used as antimicrobial, insecticidal, anti-inflammatory, and antidiabetic activities, etc. [4,5,6,7]. D-borneol, a time-honored drug in traditional Chinese medicine for more than 2000 years, for example, is extracted from Ca. *officinarum* and *Ci. burmanni* [8,9].

D-Borneol can be applied either through the skin or taken orally, with the skin application being more common [10]. It is mainly used on the skin to ease pain from wounds, injuries, burns, cuts, and the like. When taken by mouth, D-Borneol is widely used to treat cardiovascular diseases like stroke, coronary heart disease, and angina pectoris. It is a crucial ingredient in many traditional Chinese herbal remedies, such as the Angong Niuhuang pill, Suxiao Jiuxin pill, and Fufang Danshen pill (or Compound Danshen) [11,12,13,14].

Currently, there are two methods for obtaining borneol: artificial synthesis and natural extraction [15]. Synthetic borneol, which is derived chemically from turpentine oil or camphor, is inherently impure. It contains unforeseen byproducts like levogyral borneol and optically inactive isoborneol [16,17]. Apart from these byproducts, there are concerns about potential harm to the human body due to residual raw materials remaining from the chemical reaction used in borneol production, such as camphor [18]. The earliest record for extracting natural D-borneol is from *Dryobalanops aromatica* C. F. Gaertn (Dipterocarpaceae) in Southeast Asia [19]. In the 1980s, researchers in China discovered chemotypes of Ca. *officinarum* and *Ci. burmanni* with high proportions of borneol [20,21]. Subsequently, these have gradually replaced imported natural borneol from Southeast Asia and have become the primary contributors to the D-borneol production. However, *Ci. burmanni* is considered as a more ideal species because, unlike Ca. *officinarum*, D-borneol extracted from *Ci. burmanni* is not concomitant with safroles, which may affect the quality [8].

In Guangdong Province, South China, the D-borneol-rich chemotype of *Cinnamomum burmanni* is known as the Meipian tree (梅片树). Previous studies indicate that the D-borneol content in the Meipian tree ranges from approximately 19.68% to 78.6% [8]. Additionally, Wu et al. discovered two other chemotypes: cymene type and cineol type, along with two transitional types: cymene/cineol type and cineol/borneol type [20]. They proposed an evolutionary trait for these chemotypes: cymene type-cymene/cineol type-cineol type-cineol/borneol type-borneol type.

According to transcriptome studies, the biosynthesis pathway of Glycerol 3-Phosphatase (GPP), the precursor of D-borneol in vivo, is linked to both the Mevalonate (MVA) and Methylerythritol 4-phosphate (MEP) pathways [22,23,24,25]. Ma et al. used transcriptomic data to identify a key gene (CbTPS-1) in the Terpene Synthase (TPS) gene family, crucial for the downstream transformation of GPP generated by upstream pathways [25]. Although the plastid genome of *Ci. burmanni* has been reported, it was based on only a few individuals with enigmatic chemotypes [26,27].

During the development and utilization of the Meipian tree, researchers have observed significant variations in the essential oil and D-borneol content among different individuals [8,15,20,24]. Moreover, progeny from high-borneol-content mother plants also demonstrate variations in D-borneol production. As a result, cutting propagation has become a widely adopted method for mass production. However, ongoing efforts to selectively breed young, high-yield mother plants for providing cuttings and introducing genetic diversity from the wild are considered necessary to enhance the production of D-borneol. Meanwhile, botanical gardens, as crucial institutions for the relocation and conservation of wild plants, need to continually advance scientific research on wild plants and promote the sustainable utilization of resources [28,29]. Therefore, to assess the current status of living collections of *Ci. burmanni* in botanical gardens in southern China, a survey was conducted. The study reports the diversity of essential oils and the chloroplast genomic diversity of chemotypes in *Ci. burmanni*, offering valuable information for the selection and utilization of potential high-borneol-content mother plants.

## 2. Results

### 2.1. Chemical Compositions and Chemotype Identification

In total, there were 24 components in the samples with relative contents exceeding 3%, including 8 monoterpenoids, 6 sesquiterpenoids, and 7 phenylpropanoids (Table 1). Most of the samples had one or two dominant compounds, which were eucalyptol, borneol, coumarin, caryophyllene, and bicyclogermacrene. In 11 out of 28 samples, eucalyptol made up the largest proportion in essential oils (range from 20.48% to 35.85%). Eight samples were predominantly made up of borneol (range from 23.88% to 54.89%). Two samples were predominantly coumarin (29.56% and 34.31%). We also found in two samples that the main components of essential oils were methyl cinnamate (44.25%) and bicyclogermacrene (42.57%), respectively. The remaining samples did not have predominant components. In addition, there were samples with relative contents of sabinene, caryophyllene, and α-phellandrene exceeding 10%. In general, sesquiterpenoids were relatively low in content, with monoterpenoids (represented by eucalyptol) and phenylpropanoids (represented by borneol) being predominant.

Considering practical needs and the chemotype classification of Wu et al. [20], samples could be classified as eucalyptol type (11 samples), borneol type (9 samples), other type (4 samples), and mixed type (4 samples).

### 2.2. PCA and Correlation Analysis

Four principal components (PCs) accounted for 81.15% of the total variance, with PC1 accounting for 51.39% and other PCs accounting for around 10%. A strongly positive correlation was observed between PC1 and borneol with its derivative (Table 2). Meanwhile, PC1 showed a negative correlation with most of the components, including eucalyptol and α-phellandrene (Figure 1). PC2 showed positive correlations with eucalyptol and α-phellandrene, and negative correlations with coumarin, caryophyllene, etc. PC3 showed a positive correlation with coumarin and a negative correlation with bicyclogermacrene. PC4 showed positive correlations with α-phellandrene and methyl cinnamate, and negative correlations with eucalyptol and coumarin.

Correlation coefficients indicated significant positive or negative correlations among several sets of compounds. Borneol presented positive correlations with α-pinene, β-Pinene, camphor, borneol acetate, and limonene, which indicate as one group (Figure 2). Eucalyptol presented positive correlations with guaiol, sabinene, α-terpineol, cinnamaldehyde, cinnamyl acetate, methyl cinnamate, and α-phellandrene. Notably, strongly negative correlations were observed among compounds from distinct groups.

### 2.3. Plastid Genome De Novo Assembly and Gene Organization

Plastome size among the 28 newly sequenced *Ci. burmanni* in this study ranged from 152,763 bp to 152,775 bp, with a similarity of 99.1% (Figure 3). The length of the inverted repeat region (IR) ranged from 19,977 bp to 20,092 bp. The length of the large single copy region (LSC) was 93,688 bp, whereas the small single copy region (SSC) was 18,903 bp. A total of 120 genes were annotated, with 34 of them being tRNA genes.

### 2.4. Phylogenetic Analysis

The phylogenetic tree revealed three major clades of sampled *Ci. Burmanni* (Figure 4). Clade I comprised three individuals, all from ZSSMY, which was the sister group to the rest of the samples with a full support (BS = 100). In Clade II, samples from the same collection site did not cluster together. TUTY5714 was identified as the early diverging lineage with full support (BS = 100) and TUTY5713 was closely related to LKY06 (BS = 100). Clade III was the largest branch, containing samples from four collection sites. However, due to the high sequence similarity, the resolution of this branch is quite low.

### 2.5. Phylogenetic Signal Test

Compounds with Pagel’s λ values close to 1 indicate strong phylogenetic signal, suggesting that their evolution is highly correlated with the phylogeny. Compounds with low Pagel’s λ values (close to 0) have weak phylogenetic signals, suggesting that their evolution may be less constrained by phylogeny. Compounds with *p*-values < 0.05 are considered to have significant phylogenetic signals, while those > 0.05 are not. In our tests, all the compounds showed both low Page’s λ values and non-significant *p*-values, suggesting weak phylogenetic signals and no significant evolutionary correlations with the phylogeny (Table 3).

## 3. Discussion

Nearly all species within *Cinnamomum* and *Camphora* contain essential oils with diverse chemotypes. According to previous reports, the linalool type has been identified in *Ci. osmophloeum* Kaneh., *Ca*. *parthenoxylon* (Jack) Nees, *Ci. kanahirae* Hayata, *Ca*. *officinarum*, and *Ci. verum* [30,31,32,33]; the eugenol type is present in *Ci. impressinervium* Meisn. and *Ci. verum* [34,35]; the eucalyptol type is found in *Ci. kanahirae*, Ca. *parthenoxylon*, and *Ca. officinarum* [31,32]; while the borneol type is exclusive to *Ci. camphora* and *Ci. burmanni* [9,20]. Other chemotypes, including safrole, cinnamaldehyde, citral, camphor, and nerolidol types, have also been identified in various *Cinnamomum* s.l. species [31,36,37].

Concerning the collections in the botanical gardens of Guangdong Province in southern China, *Ci. burmanni* primarily exhibits two chemotypes: eucalyptol type and borneol type. Notably, individuals such as LKY03, LKY06, LKY07, and TUTY5713 yield borneol at a rate of more than 50%. Two individuals, LKY01 and LKY 05, with a high percentage of coumarin were also discovered. 

Coumarin, chemically known as 2H-chromen-2-one, is a compound that is widespread in plants, encompassing various vegetables, spices, fruits, and medicinal plants [38]. This compound has been used in several countries for the treatment of conditions such as edemas, renal cell carcinoma, and other tumors [39]. However, evidence of the hepatotoxic effects and possible carcinogenicity in rodent experiments has been reported [40,41]. Until a medical consensus is established with regard to coumarin, botanical gardens have a responsibility to safeguard these two individuals and create opportunities for the exploration of potential new drugs [42].

Compounds that show a positive correlation may be synthesized by the same enzyme or by different enzymes in the same synthetic pathway. On the other hand, negatively correlated chemicals may indicate a substrate competition relationship or an upstream and downstream relationship within the same synthetic route. In previous research, seven functional genes (CbTPS1 to CbTPS7) in the terpene synthase (TPS) gene family were found to be linked to the synthesis of borneol [25]. Additionally, components positively correlated with borneol are regulated by these genes. Our correlation analysis suggests a negative correlation between borneol and eucalyptol, indicating a substrate competition relationship or an upstream and downstream connection within the same synthetic route. Identifying relevant genes and regulating them may help screen for individuals with higher borneol content and lower eucalyptol.

The three clades of the phylogenetic tree exhibited different chemotypes (Figure 4). In Clade I, two other types and one mixed type were identified, and methyl cinnamate, α-phellandrene, and (2E)-2-hexene, respectively, were the compounds with the highest content. In Cade II, all samples were borneol type, with the content ranging from 44.08% to 54.89%. Samples from Clade III were mostly eucalyptol type, with a few borneol, other, and mixed types. Due to all the samples from Clade II containing high-content borneol, there may be a relationship between synthesis of borneol and maternal inheritance. Nevertheless, phylogenetic signal tests revealed low Pagel’s λ values and non-significant *p*-values for these compounds, indicating weak phylogenetic signals and no significant evolutionary correlations with the phylogeny (Table 3). 

Our research has also brought to light that, historically, botanical gardens lacked a clear collection purpose, often concentrating solely on the different species. In situations where spatial resources are limited and extensive collection and cultivation are not feasible, a more targeted approach to botanical garden resource collection becomes imperative. Specifically, for plant families like Lauraceae and Lamiaceae, where essential oils constitute the primary utilizable components, collection strategies should be informed by an understanding of chemotypes. This approach could be more effective in both collecting and preserving the genetic diversity and chemical component diversity of species.

For future research, on the one hand, individuals in Clade II (LKY03, LKY06, LKY07, and TUTY5713) could be high-yield mother plants for providing cuttings. On the other hand, outcrossing among four individuals could be conducted to examine whether the high expression of borneol in their offspring is stable. If not, searching for offspring with high and low expression of borneol by sequencing their transcriptomes would aim to identify more genes associated with GPP and validate previously identified genes related to borneol synthesis.

## 4. Materials and Methods

### 4.1. Plant Materials

The 28 studied *Cinnamomum burmanni* samples were collected in Guangdong Academy of Forestry (GAF, E 113°38′ N 23°20′), South China National Botanical Garden (SCNBG, E 113°37′ N 23°18′), Foshan Botanical Garden (FBG, E 113°00′ N 23°10′) and Zhongshan Arboretum (ZA, E 113°37′ N 22°49′). Voucher specimens were deposited in the IBCS (Table 1). Two leaf tissue samples from each individual were collected for volatile terpenoid and DNA extraction. For DNA extraction, fresh leaves were preserved in silica-gel; for volatile terpenoid extraction, mature leaves were (SPAD ≥ 35) stored in a cool dry container and were extracted in 24 h.

### 4.2. Essential Oils Extraction and Identification

Four grams of leaves of each sample were put in stainless steel tubes individually, and immersed the tubes into liquid nitrogen for 5 min, then the leaves were ground into powder by SCIENTZ-48 (Scientz City, China). Next, 2 g of powder was added into centrifuge tubes and extracted with 4 mL hexane using the ultrasonic cleaner for 30 min, and then incubated at 56 °C for 1 h. Samples were centrifuged at 10,000 rpm for 5 min and the supernatants were pipetted into new centrifuge tubes. Following this, 0.5 μL samples of the supernatants were pipetted into vials for GC-MS analysis using Shimadzu GCMS-QP2020 (Shimadzu City, Japan). Helium was employed as the carrier gas (30 mL/min), and the volatile compounds separated on the SH-Rxi-5Sil MS column (30 m × 0.25 mm × 0.25 μm) with the inlet heated at 230 °C. GC oven temperature was initiated at 70 °C, with an Increase of 2 °C/min to 160 °C, and kept at 160 °C for 2 min, and then climbed to 220 °C at 10 °C/min, with a final hold at 220 °C for 5 min. The GC-MS interface and ionization source temperature were 250 °C and 200 °C, respectively. The essential oils were identified both by NIST05 and their retention index. The relative concentrations of essential oils were determined by their chromatographic peak area using a normalization method.

### 4.3. Statistical Analysis of Essential Oils

The chemical compositions of the leaf samples with a percentage above 3% were used as variables in the analyses. Firstly, standardization of the data matrix was conducted by subtracting the mean and dividing it by the standard deviation. The principal component analysis and correlation coefficient analysis were plotted in R [43] using packages ggplot2 [44], ggforce [45], and corrplot [46] to verify the interrelation in the oil’s components. Phylogenetic signal texts were also conducted in R using packages tidyverse [47], ape [48], geiger [49], phytools [50], and caper [51].

### 4.4. DNA Extraction, Sequencing, Plastome Assembly, and Annotation

Whole genomic DNA was isolated from leaf tissue dried in silica gel following the modified CTAB protocol [52]. We fragmented the isolated total genomic DNA into ca. 300–500 bp in length to construct a library following the manufacturer’s manual (Illumina). Then, DNA fragments were end-polished, A-tailed, and ligated with the full-length adapter for Illumina sequencing. The DNA libraries were sequenced on the Illumina HiSeq X-Ten instrument at Beijing Genomics Institute (BGI) or Novogene Bioinformatics Institute, and 150 bp paired-end reads were generated. Finally, approximate 3–5 Gb high-quality sequences were obtained for each sample. We used GetOrganelle pipeline v.1.7.5 [53], SPAdes [54], and Bandage v.0.8.1 [55] to assemble and visualize the plastomes. We employed Geneious Prime v.2019–v.2021.2.2 [56] to verify the accuracy of the assembly and to annotate the plastome. The annotated plastomes were deposited in GenBank (Table 4).

### 4.5. Phylogenetic Analysis

To reconstruct the phylogenetic relationships, we generated a data matrix using 28 newly sequenced plastomes of *Cinnamomum burmanni* and 1 previously published plastid genome as outgroups from GenBank. The whole plastid genomes were aligned by using MAFFT v7 [57]. We used IQ-TREE v1.6.12 [58] to reconstruct the best ML tree. The best-fit models were recommended by ModleFinder v2.2.0 [59]. The branch supports were estimated using 1000 interactions of standard bootstrap (-b 1000). Bootstrap percentage (BP) values of 90% or higher were considered statistically significant and indicated a well-supported clade, while those with 70–89% and 50–69% corresponded to a moderately and a weakly supported clade, respectively. Trees with stacked bar plots showing essential oil components above 4% were visualized by Chiplot [60].

## Figures and Tables

**Figure 1 molecules-29-01206-f001:**
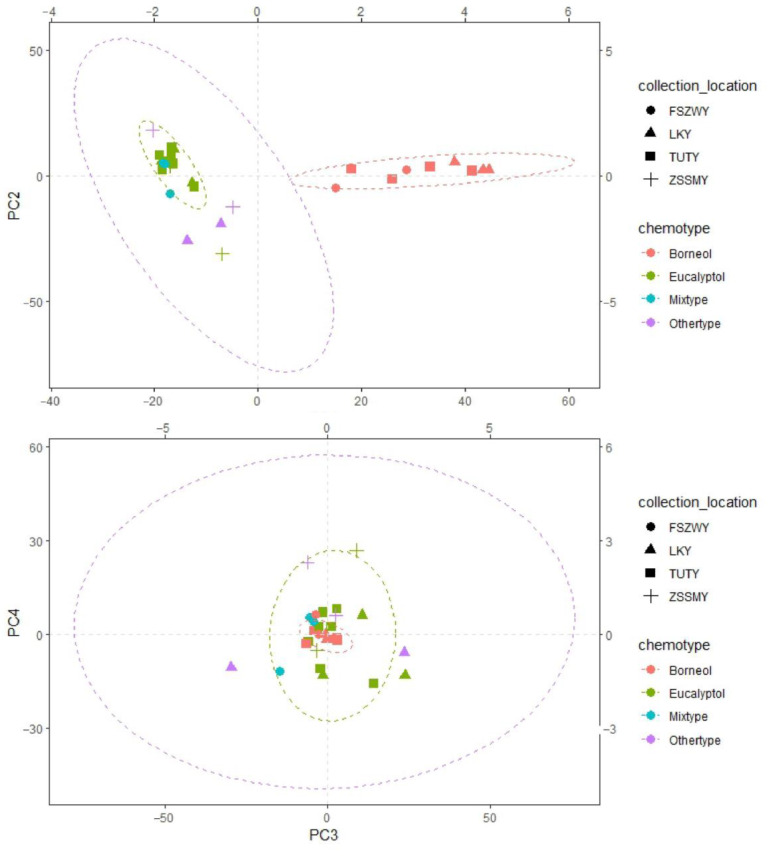
Principal component analysis of the first four dimensions.

**Figure 2 molecules-29-01206-f002:**
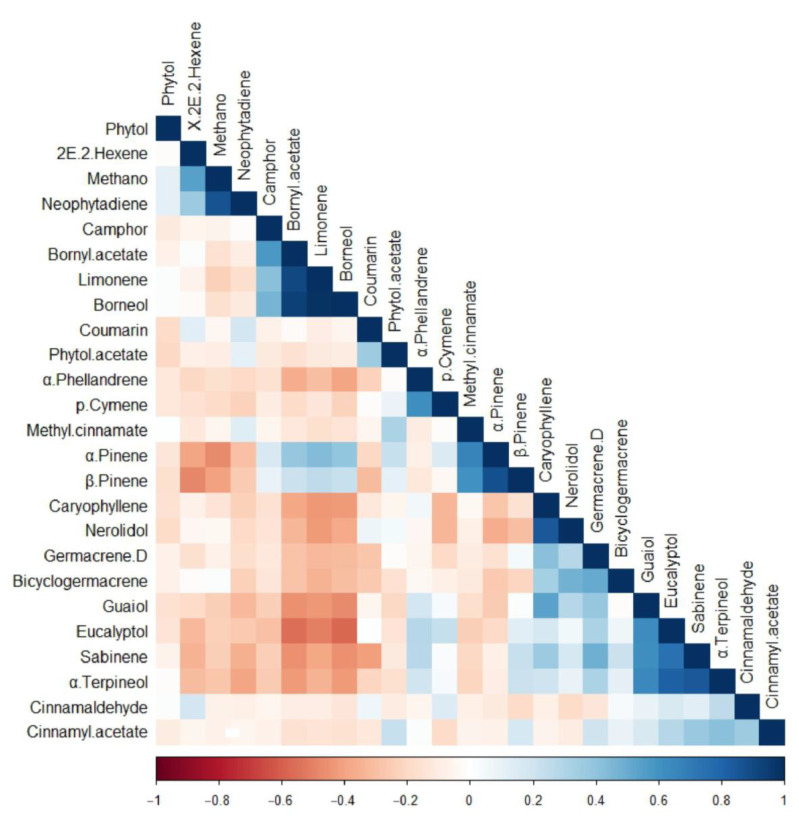
Correlation analysis of the 25 compounds identified in the essential oil of *Ci. burmanni*.

**Figure 3 molecules-29-01206-f003:**
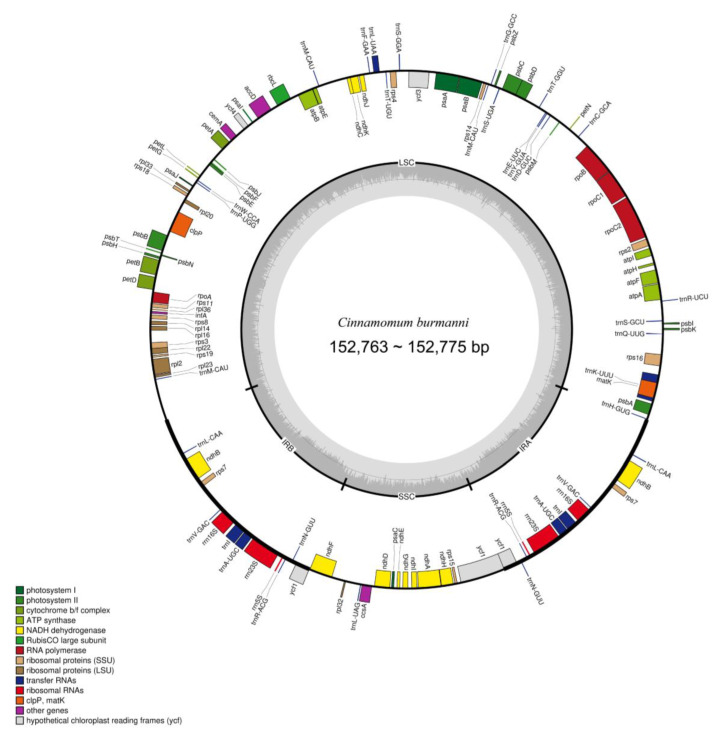
Gene map of the plastid genome of *Cinnamomum burmanni*.

**Figure 4 molecules-29-01206-f004:**
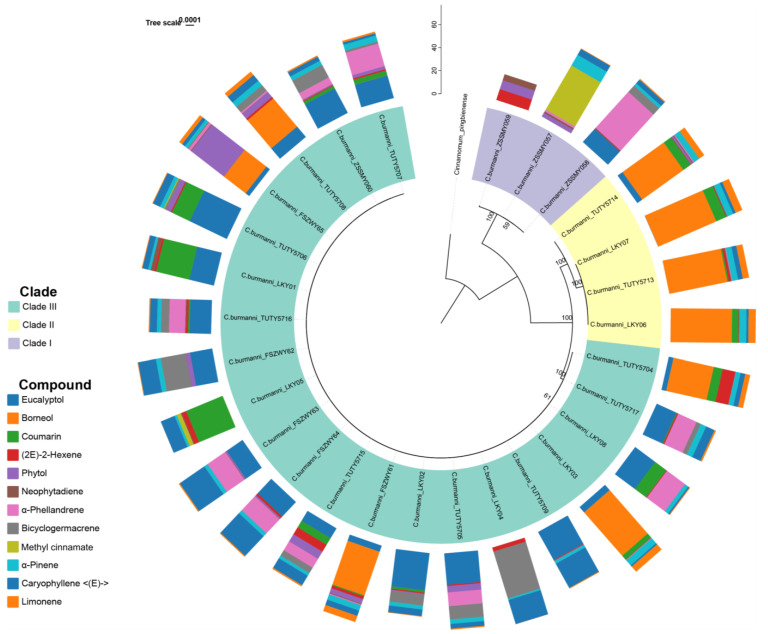
Maximum likelihood phylogram of plastome sequences from 28 *Cinnamomum burmanni* with bars showing relative content of 12 compounds for each sample. Outgroup is rooted.

**Table 1 molecules-29-01206-t001:** Relative content of 24 compounds in essential oil and samples with different chemotypes.

Compounds	RI	LKY03	LKY05	TUTY5706	TUTY5713	ZSSMY057	ZSSMY060	FSZWY61	FSZWY63
(2E)-2-Hexene	850	0.32	4.12	1.07	1.88		0.67	2.45	
α-Pinene	933	5.78	2.1	3.47	5.06	10.69	4.04	4.8	3.43
Sabinene	972	1.92		2.1			6.92	1.48	6.21
β-Pinene	978	2.15		1.97	1.81	4.44	1.75	1.87	1.41
α-Phellandrene	1007	0.55			0.65	1.89	6.52	0.93	18.69
p-Cymene	1025	2.34		0.89	0.53	2.3	6.58		0.41
Limonene	1030	5.88		0.59	5.48	0.42	1.1	5.6	0.94
Eucalyptol	1032	6.05		35.85			24.01	5.65	16.8
Camphor	1149				4.13				
Borneol	1173	50.15			51.49			40.15	
α-Terpineol	1199	2.26	0.9	7.37			6.26	1.53	3.92
(E)-Cinnamaldehyde	1273	0.03					1.08		
Bornyl acetate	1285	2.81			6.89			2.82	
Methyl cinnamate	1384		3.8	0.82		44.25			
(E)-Caryophyllene	1424	1.33	12.86	6.64	4.22	5.87	4.01	4.85	25.9
Coumarin	1438	4.47	34.31	17.06	1.24		2.69	1.01	
(E)-Cinnamyl acetate	1448							0.93	
Bicyclogermacrene	1497			1.9			12.41		
(E)-Nerolidol	1561		2.51	1.29					3.2
Germacrene D	1480								
Guaiol	1603		2.3	2.21			2.81	1.44	2.05
Neophytadiene	1836		0.86	0.16	0.6	1.46	0.26	0.71	0.39
Phytol	2106			4.1		3.54	0.75	4.05	1.05
Phytol acetate	2212	4.66	6.77			4.26			

**Table 2 molecules-29-01206-t002:** Four principal components of the 25 essential oil compounds of *Ci. burmanni*.

Compounds	PC1	PC2	PC3	PC4
(2E)-2-Hexene	0.007137	−0.07748	0.025247	−0.01351
α-Pinene	0.03166	−0.01109	0.013891	0.079372
Sabinene	−0.0747	0.120587	−0.10141	−0.06655
β-Pinene	0.007353	0.004175	0.001428	0.020175
α-Phellandrene	−0.19855	0.507336	−0.10965	0.618629
p-Cymene	−0.0315	0.110504	0.037194	0.122443
Limonene	0.089563	0.047327	−0.01443	0.002037
Eucalyptol	−0.31716	0.54983	0.097892	−0.39991
Camphor	0.015634	0.004143	−0.00539	0.001487
Borneol	0.889358	0.249417	−0.12253	−0.0748
α-Terpineol	−0.05399	0.104253	−0.03002	−0.07722
(E)-Cinnamaldehyde	−0.01036	0.007017	0.013644	−0.02766
Bornyl acetate	0.100396	0.026362	−0.01025	−0.00878
Methyl Cinnamate	−0.02336	−0.45327	0.17169	0.442389
(E)-Caryophyllene	−0.14359	−0.15773	−0.25804	−0.16853
Coumarin	−0.00125	−0.12113	0.673163	−0.315
Methano	−0.00269	−0.02349	0.002636	0.00352
(E)-Cinnamyl acetate	−0.0114	0.019374	−0.00923	−0.01975
Germacrene D	−0.02679	−0.01251	−0.07591	−0.05683
Bicyclogermacrene	−0.12734	−0.26582	−0.61817	−0.27916
(E)-Nerolidol	−0.01728	−0.03422	−0.03042	−0.03499
Guaiol	−0.03271	0.036051	−0.00172	−0.03787
Neophytadiene	−0.00028	−0.03547	0.037002	0.014107
Phytol acetate	−0.00544	−0.05115	0.084152	0.034344
Phytol	0.018862	−0.09068	−0.05362	0.098565

**Table 3 molecules-29-01206-t003:** Phylogenetic signal tests of essential oils in *Cinnamomum burmanni*.

Compounds	Pagel’s λ	*p*-Value
Eucalyptol	0.52	0.15
Borneol	0.66	0.19
Coumarin	0.28	0.44
(2E)-2-Hexene	7.33 × 10^−5^	1
Phytol	7.33 × 10^−5^	1
Neophytadiene	7.33 × 10^−5^	1
α-Phellandrene	7.33 × 10^−5^	1
Bicyclogermacrene	7.33 × 10^−5^	1
Methyl cinnamate	7.33 × 10^−5^	1
α-Pinene	0.19	7.33 × 10^−15^
Caryophyllene	0.19	1.02 × 10^−10^
Limonene	7.33 × 10^−5^	1

**Table 4 molecules-29-01206-t004:** Voucher specimens of *C. burmanni* associated with sampling.

Voucher Number	Collecting Location	Collecting Date	Collector	Tree Height (m)	Diameter at Breast Height (cm)
LKY01	GA	2022.08.03	PW Xie, JL Chen, JC Zhan	4.6	12.5
LKY02	GA	2022.08.03	PW Xie, JL Chen, JC Zhan	3.8	10.5
LKY03	GA	2022.08.03	PW Xie, JL Chen, JC Zhan	7.5	17.8
LKY04	GA	2022.08.03	PW Xie, JL Chen, JC Zhan	3.3	10.2
LKY05	GA	2022.08.03	PW Xie, JL Chen, JC Zhan	3.8	10.6
LKY06	GA	2022.08.03	PW Xie, JL Chen, JC Zhan	6.8	16.5
LKY07	FBG	2022.08.03	PW Xie, JL Chen, JC Zhan	7.2	14.2
LKY08	FBG	2022.08.03	PW Xie, JL Chen, JC Zhan	7.8	21.8
FSZWY61	FBG	2022.08.07	TY Tu, PW Xie	8.8	31.8
FSZWY62	FBG	2022.08.07	TY Tu, PW Xie	8.5	33.5
FSZWY63	FBG	2022.08.07	TY Tu, PW Xie	8.5	30.5
FSZWY64	FBG	2022.08.07	TY Tu, PW Xie	7.8	31.2
FSZWY65	FBG	2022.08.07	TY Tu, PW Xie	7.5	28.8
TUTY5704	FBG	2022.08.07	TY Tu, PW Xie	3.2	10.5
TUTY5705	FBG	2022.08.07	TY Tu, PW Xie	2.5	8.2
TUTY5706	FBG	2022.08.07	TY Tu, PW Xie	2.2	7.8
TUTY5707	FBG	2022.08.07	TY Tu, PW Xie	7.0	20.6
TUTY5708	FBG	2022.08.07	TY Tu, PW Xie	7.5	22.4
TUTY5709	FBG	2022.08.07	TY Tu, PW Xie	7.2	21.5
TUTY5713	SCNBG	2022.08.17	TY Tu	7.6	20.2
TUTY5714	SCNBG	2022.08.17	TY Tu	2.1	7.5
TUTY5715	SCNBG	2022.08.17	TY Tu	5.8	15.8
TUTY5716	SCNBG	2022.08.17	TY Tu	4.5	12.6
TUTY5717	SCNBG	2022.08.17	TY Tu	7.5	22.6
ZSSMY057	ZA	2022.08.15	HM Lian, YL Zhong, PW Xie	5.2	15.6
ZSSMY058	ZA	2022.08.15	HM Lian, YL Zhong, PW Xie	5.7	15.8
ZSSMY059	ZA	2022.08.15	HM Lian, YL Zhong, PW Xie	5.7	16.2
ZSSMY059	ZA	2022.08.15	HM Lian, YL Zhong, PW Xie	4.5	16.5

Abbreviations of collecting location: GA: Guangdong Arboretum; FBG: Foshan Botanical Garden; SCNBG: South China National Botanical Garden; ZA: Zhongshan Arboretum. Abbreviations of deposit site: GAF: Guangdong Academy of Forestry; IBSC: Herbarium, South China National Botanical Garden, Chinese Academy of Sciences.

## Data Availability

All the data are shown in the main manuscript.

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
