# Peer review of "Unpredictable Chemical Diversity of Essential Oils in Cinnamomum burmanni (Lauraceae) Living Collections: Beyond Maternally Inherited Phylogenetic Relationships"

_molecules, 2024, doi:10.3390/molecules29061206_

Round 1
Reviewer 1 Report
Comments and Suggestions for Authors
The manuscript "Phylogenetic and Chemical Diversity of Essential Oil in Cinnamomum burmanni (Lauraceae) from Living Collections of Botanical Gardens" is interesting, but the material used in the study is insufficiently described.
Please provide legend for table 2.
Please take care, table 1 is in material and method (it should be table 4)! In my opinion it would be very interesting to give some specifics for each voucher specimens. Or pictures.
It is big challenge to study a botanical collection, so I think more information (morphologicaly in first line) should be given about the examined plant material.
Author Response
1) With over a decade of dedicated research into Cinnamomum burmanni, our team is very familiar with this species.
2) C. burmanni has unique morphological characteristics that make it easy to identify. While C. japonicum shares morphological and phylogenetic similarities, notable differences include the perianth cup in fruit of C. burmanni, which regularly exhibits dentate margins with teeth that are truncate at the apex, whereas C. japonicum typically features a perianth cup in fruit that is entire or shallowly dentate on margin. However, we do not think it is necessary to provide morphological details of this species, as the focus of this paper lies primarily on essential oil composition and plastid genome analysis.
3) Through our phylogenetic analyses, we’ve observed minimal variance in the chloroplast genome among these samples, with a notable difference from the closest outgroup. As such, species identification for all samples should pose no significant challenges.
4) Number of tables have been revised. So as the legend of table 2.
Reviewer 2 Report
Comments and Suggestions for Authors
1. It is necessary to investigate the plant's introduction records in detail, and give a detailed account of the introduction of all 28 "æ ªï¼Ÿ"trees, especially information about the age, place of introduction, and method of introduction (i.e. immediate harvesting/seed source/family line/single plant/excellent plant, etc.), etc., which affects the composition of the essential oils and their components.
2. It is not recommended to use "Phylogenetic..." for the analysis of chemotypic diversity within the species.
The expression "The genus Cinnamomum s.l. consists of two segregated genera (Camphora and Cin- 33 namomum s.s.)" is questionable and raises doubts about the accuracy of its species identification. The knowledge of the species identification is questionable.
The collection of samples for essential oil extraction, the treatment and extraction method, the adsorption column of the instrument, and the reproducibility must be carefully accounted for, which is the key to the correctness of the conclusions.
Comments on the Quality of English LanguageEliminate Chinese English as much as possible.
Author Response
1) With over a decade of dedicated research into Cinnamomum burmanni, our team is very familiar with this species.
2) C. burmanni has unique morphological characteristics that make it easy to identify. While C. japonicum shares morphological and phylogenetic similarities, notable differences include the perianth cup in fruit of C. burmanni, which regularly exhibits dentate margins with teeth that are truncate at the apex, whereas C. japonicum typically features a perianth cup in fruit that is entire or shallowly dentate on margin.
3) Through our phylogenetic analyses, we’ve observed minimal variance in the chloroplast genome among these samples, with a notable difference from the closest outgroup. As such, species identification for all samples should pose no significant challenges.
4) Title has been revised.
Round 2
Reviewer 1 Report
Comments and Suggestions for Authors
In my opinion, "Maternal" should be removed from the title, and it should be reconstructed: "Phylogeny Does Not Predict Chemical Diversity of Indonesian Cassia (Cinnamomum burmanni, Lauraceae) Essential Oil"
I suggest that the manuscript be accepted after correcting the title in its current form.
Author Response
- Title has been revised as "Chemical Diversity of Essential Oil in living collections of Cinnamomum burmanni (Lauraceae) from botanical gardens can not be predicted by maternally inhereted phylogenetic relationships“
- Add Phylogenetic signal tests
Reviewer 2 Report
Comments and Suggestions for Authors
1. For the initial collection of seed sources, there was no particularly in-depth analysis of the chemical types of germplasm, and the depth of research was insufficient.
2. The revised title has added uncertainty to the research conclusion.
Comments on the Quality of English LanguageStrengthen discussions and avoid Chinglish.
Author Response

(The authors gave the same response as above.)
